# Pleural Mesothelioma Diagnosis for the Pulmonologist: Steps Along the Way

**DOI:** 10.3390/cancers17233866

**Published:** 2025-12-01

**Authors:** Alberto Fantin, Nadia Castaldo, Ernesto Crisafulli, Giulia Sartori, Filippo Patrucco, Horiana B. Grosu, Paolo Vailati, Giuseppe Morana, Vincenzo Patruno, Stefano Kette, Avinash Aujayeb, Aleš Rozman

**Affiliations:** 1Department of Pulmonology, S. Maria della Misericordia University Hospital, 33100 Udine, Italy; 2Respiratory Medicine Unit, Department of Medicine, University of Verona, Azienda Ospedaliera Universitaria Integrata of Verona, 37139 Verona, Italy; 3Division of Respiratory Diseases, Department of Medicine, Maggiore della Carità University Hospital, 28100 Novara, Italy; 4Department of Pulmonary Medicine, The University of Texas MD Anderson Cancer Center, Houston, TX 77030, USA; 5Pulmonology Unit, Department of Medical Surgical and Health Sciences, University of Trieste, Hospital of Cattinara, 34149 Trieste, Italy; 6Respiratory Department, Northumbria Healthcare NHS Foundation Trust, North Shields NE29 8NH, UK; 7Faculty of Medicine, University of Ljubljana, 1000 Ljubljana, Slovenia; 8University Clinic of Respiratory and Allergic Diseases, 4204 Golnik, Slovenia

**Keywords:** pleura, mesothelioma, diagnosis, imaging, biopsy, thoracoscopy, biomarkers

## Abstract

This review explains how pulmonologists guide patients through the complex diagnostic process of malignant pleural mesothelioma, from initial imaging tests to obtaining tissue samples and confirming the diagnosis. It also describes how new technologies may improve accuracy and reduce the need for invasive procedures. By summarizing the most up-to-date evidence, this paper aims to support more precise diagnosis and better communication between healthcare professionals, patients, and families. We also highlight the importance of multidisciplinary collaboration and innovation in improving care for people affected by mesothelioma.

## 1. Introduction

### 1.1. Background and Rationale

Malignant pleural mesothelioma (MPM) is a rare and aggressive neoplasm that arises from the mesothelial cells lining the pleural cavity. Despite advances in oncological care, prognosis remains poor, with median survival rarely exceeding one year after diagnosis [1]. 

Early and accurate diagnosis of MPM is crucial, as it influences therapeutic decision-making, eligibility for multimodal treatment, and ultimately patient outcomes [2]. However, the diagnostic pathway is frequently challenging due to the non-specificity of symptoms [3], which overlap with a variety of benign and malignant conditions, or the absence of known occupational exposure to risk factors [4]. In addition, non-invasive diagnostic tools, such as imaging modalities, are often limited by constraints on sensitivity and specificity.

In recent years, progress has been made in refining diagnostic strategies through improved imaging protocols, advances in endoscopic techniques, the application of immunohistochemistry panels, and the investigation of novel molecular and circulating biomarkers [5]. Furthermore, international guidelines have sought to standardize diagnostic approaches [6], although variations in clinical practice and access to resources persist.

Given the complexity and heterogeneity of diagnostic options, there is a pressing need to synthesize available evidence and provide a comprehensive overview tailored to pulmonologists, who are frequently the first specialists to encounter suspected cases. 

### 1.2. Aims and Scope of the Review

To synthesize the diagnostic pathways available to patients, as managed under a pulmonologist’s guidance. The review will examine the clinical reasoning and decision-making process in selecting appropriate investigations and discuss the integration of conventional and emerging diagnostic modalities. Attention is also paid to the patient’s experience during the communication of the diagnosis.

## 2. Methodology of the Review

This review evaluates the existing literature by collecting primary English-language bibliographic references from international scientific databases (PubMed and Scopus) using the search terms “mesothelioma” or “pleural mesothelioma”. The search strategy aimed to include the most significant documents for the human species, dealing with the diagnostic pathway of MPM. The report covered the period from January 2005 to June 2025. Systematic reviews, meta-analyses, randomized controlled trials (RCTs), original research papers and case reports were included in our initial search (see Appendix A for the complete search strategy; Appendix A). We included human studies in the adult population. The authors included other references considered significant.

Following completion of the literature searches, all retrieved records were imported into the Evidence Review Accelerator (https://tera-tools.com/, accessed on 1 June 2025), which facilitated identification and removal of both exact and near-duplicate entries. Subsequently, two reviewers independently screened titles and abstracts to identify eligible documents. Full-text articles deemed potentially pertinent were evaluated in the same independent fashion. Discrepancies between reviewers were addressed through structured discussion, with recourse to a third reviewer when consensus could not be reached. 

Of note, the list of references included is not necessarily all-encompassing but reflects the body of evidence believed appropriate to the purpose of this document: highlighting the latest progress in the diagnostic pathway of MPM.

## 3. Epidemiology, Etiology and Pathogenesis

### 3.1. Occupational and Environmental Risk Factors

Occupational exposure to asbestos fibers remains the principal etiological factor for MPM [7]. Historically, individuals employed in the shipbuilding, construction, insulation, mining, and manufacturing industries have been disproportionately affected by prolonged, often unprotected exposure to asbestos-containing materials [8,9,10,11,12,13]. Secondary occupational exposure, such as that encountered by family members of workers through contaminated clothing, has also been documented and contributes to the overall disease burden [14]. The male-to-female ratio for MPM prevalence is usually 4:1, with possible significant interregional variations worldwide [15].

Environmental exposure represents an additional source of risk, particularly in regions where asbestos deposits naturally occur or where improper disposal of asbestos-containing materials has contaminated soil and air [7]. Residential proximity to asbestos mines or industries that process asbestos has been linked to higher incidence rates, underscoring the significance of community-level exposure [7].

Although asbestos remains the predominant risk factor, other potential contributors have been explored, including exposure to erionite fibers [16], ionizing radiation [17,18], and certain genetic predispositions identified in familial clusters [19,20,21]. These factors, while less common, may interact synergistically with asbestos exposure to accelerate disease onset or progression [22,23]. In particular, localized outbreaks of MPM due to exposure to erionite fibers in families with a genetic predisposition to oncogenesis have been described [24,25,26].

Recognition of both occupational and environmental risks is essential for diagnosis, as a thorough exposure history remains a cornerstone of clinical assessment. Despite this, the long latency period between exposure and clinical manifestation, which often spans decades, frequently complicates timely recognition [27,28].

The diagnostician who follows the patient through the process of recognizing the disease must accurately reconstruct the patient’s employment history and both occupational and residential exposure to complete their risk profile [29] and, once the disease has been identified, to initiate the process of recognizing financial compensation or a pension, as required in some states for patients with asbestos-related diseases [30].

### 3.2. Pathogenesis and Disease Progression

The pathogenesis of MPM is primarily initiated by inhalation and retention of asbestos fibers, which trigger chronic inflammation, oxidative stress, and repeated mesothelial injury. In this milieu, the nuclear protein High Mobility Group Box 1 (HMGB1) plays a central role: upon mesothelial cell damage or death, HMGB1 is released from the nucleus to the cytoplasm and into the extracellular space, where it acts as a damage-associated molecular pattern (DAMP), driving inflammatory cytokine release (such as Tumor necrosis factor-α), recruiting immune cells, and fostering a mutagenic microenvironment via reactive oxygen species [31,32,33]. Over the decades, the accumulation of genetic damage in mesothelial cells leads to malignant transformation [33].

At the molecular level, BRCA1-associated protein-1 (BAP1) emerges as a key tumor suppressor frequently inactivated in both germline and somatic settings [34]. In many cohorts, approximately 60–70% of sporadic mesotheliomas exhibit nuclear loss of BAP1, consistent with biallelic inactivation. However, reported frequencies vary depending on methodological factors, such as the sensitivity of sequencing approaches, the ability to detect larger deletions or copy-number alterations, and the interpretation of immunohistochemistry, as well as biological heterogeneity within tumors [33]. Concurrently, loss of Cyclin-Dependent Kinase Inhibitor 2A (CDKN2A), often through homozygous deletion, is common and synergizes with BAP1 loss [33]. Other tumor suppressors, such as NF2, are also frequently altered [35].

In hereditary cases (BAP1 tumor predisposition syndrome) [36], germline BAP1 mutations significantly increase susceptibility: affected individuals may present with mesothelioma at younger ages, even with minimal asbestos exposure, and their tumors tend to be less aggressive, with more prolonged survival [37]. 

Disease progression in MPM characteristically involves diffuse pleural surface spread rather than isolated masses, invasion of adjacent structures (e.g., chest wall, diaphragm, pericardium), and, in later stages, possible distant spread [38]. The slow, insidious progression is often compounded by tumor heterogeneity (molecular, histological, and immunological), which has implications for diagnostic sensitivity and therapeutic responsiveness [39,40,41,42]. 

## 4. Clinical Presentation and Initial Assessment

### 4.1. Symptoms and Signs

The most common initial symptom is progressive dyspnea [43], often resulting from the presence of a malignant pleural effusion. Chest pain, which may be pleuritic, dull, or diffuse, is another prevalent manifestation, reflecting both local inflammation and tumor infiltration of the chest wall or diaphragm [43].

Other frequently reported symptoms include persistent cough, fatigue, weight loss, and, in more advanced stages, anorexia and cachexia [44,45]. On physical examination, reduced breath sounds, dullness to percussion, and decreased chest expansion on the affected side are typical findings associated with pleural effusion or thickening.

As the disease progresses, signs of local invasion may become evident, such as chest wall deformities, palpable masses, or brachial plexus involvement leading to neuropathic pain [46]. The lymphatic structures [47] and the superior vena cava may also be affected [48], the latter by compression, infiltration or thrombosis. Some cases may manifest directly with signs and symptoms of pericardial involvement, up to cardiac tamponade [49] or constrictive pericarditis [50]. In other instances, peritoneal and distant metastasis involvement may also be present [6]. 

Given the overlap of these manifestations with benign pleural disease and other malignancies, the recognition of symptoms and signs alone is insufficient for diagnosis. Nevertheless, a detailed clinical evaluation remains an essential first step in raising suspicion of MPM, particularly in patients with a relevant history of asbestos exposure.

### 4.2. Differential Diagnosis

The differential diagnosis of MPM is broad, owing to its non-specific clinical manifestations and overlapping radiological and cytological features with other pleural diseases [51]. A key distinction must be made between MPM, reactive mesothelial hyperplasia and metastatic carcinomas involving the pleura [52,53,54]. These latter malignancies frequently present with pleural effusions and diffuse pleural thickening, thereby mimicking MPM both clinically and radiologically [55]. Other malignancies that may simulate MPM include diseases originating from local pleural and chest wall tissues, such as lymphomas [56], sarcomas [57], and rare chest wall tumors [58,59,60]. 

Benign pleural conditions also feature prominently in the differential diagnosis. Benign asbestos-related pleural effusions, pleural plaques, and diffuse pleural thickening due to different etiologies (e.g., resolved infection) can all produce symptoms and radiological appearances suggestive of MPM [61]. Pleural plaques are also frequently seen concurrently with mesothelioma, on the same or opposite side of the chest [62]. Similarly, chronic active inflammatory pleuritis (e.g., tuberculosis) [63] or storage diseases (e.g., amyloidosis) [64] and autoimmune-related pleural disease, such as Immunoglobulin Type 4 (IgG4)-related disease and rheumatoid arthritis, may closely resemble the early phases of MPM both clinically and cytologically [65,66]. Additionally, after an accurate diagnosis of non-specific pleuritis, 3.5–14% of patients will develop MPM, usually within the first year after diagnosis [67].

Given these challenges, a systematic approach is paramount, combining the patient’s history, imaging findings, histological sampling, immunohistochemistry (IHC) characterization and molecular analyses [68,69,70,71].

### 4.3. Role of the Pulmonologist in Early Evaluation

Pulmonologists frequently represent the first point of specialist contact for patients presenting with symptoms suggestive of pleural disease or imaging findings associated with MPM requiring further investigation, positioning them at the forefront of the diagnostic pathway [72]. Their role begins with the first clinical evaluation. 

In the outpatient or acute care setting, pulmonologists are responsible for the initial diagnostic work-up, typically including chest radiography, thoracic ultrasound (TUS), and computed tomography (CT) of the chest [6,73]. These imaging modalities help characterize pleural involvement, detect pleural thickening or nodularity, and guide decisions and planning on further diagnostic interventions. 

Pulmonologists also play a significant role in performing diagnostic and therapeutic pleural interventions [46]. These procedures are crucial not only for the acquisition of pleural fluid or tissue but also for long-term control of pleural effusion.

Beyond procedural aspects, pulmonologists serve as a bridge to the multidisciplinary team, coordinating with radiologists, pathologists, oncologists, radiotherapists and palliative care specialists to ensure timely interpretation of findings and prompt initiation of further diagnostic steps when needed [74,75,76]. Their ability to integrate clinical, radiological, and pathological data is fundamental in reducing diagnostic delays, which remain a significant barrier to early intervention in MPM.

## 5. Imaging Modalities in the Diagnostic Pathway

### 5.1. Chest Radiography

Chest radiography is usually the first-line imaging modality in patients with suspected pleural disease, owing to its wide availability, low cost, and rapid execution. In MPM, typical findings include a unilateral pleural effusion, the most common initial abnormality, and, less frequently, diffuse pleural thickening, nodularity, or mediastinal shift [77]. These features, however, are non-specific and can be observed in a range of pleural disorders [78].

While chest X-ray (CXR) lacks sensitivity and specificity for diagnosing MPM, it remains a valuable tool for early detection of pleural abnormalities and for prompting further investigations, such as TUS or chest CT [79]. In addition, serial radiographs can assist in monitoring effusion recurrence and assessing response to pleural interventions, such as observing complete lung expansion, or in suggesting a non-expandable lung [80].

Nonetheless, radiographic interpretation can be challenging in the presence of small effusions or subtle pleural changes, which are often missed. Therefore, CXR should be considered a first screening tool within a stepwise diagnostic pathway rather than a definitive imaging modality [77].

### 5.2. Thoracic Ultrasound

TUS is an essential adjunct in the early assessment of patients with suspected MPM [29]. Compared with CXR, TUS offers superior capability for detecting small pleural effusions and provides valuable information on sonographic characteristics, such as septations, pleural thickening, pleural nodules, and diaphragmatic involvement [81,82].

From a procedural perspective, TUS plays a pivotal role in guiding pleural interventions, including thoracentesis, percutaneous pleural biopsy, and the placement of both regular chest drains and indwelling pleural catheters. Image guidance significantly reduces the risk of complications such as pneumothorax, intercostal artery damage and organ puncture, and is therefore strongly recommended by international guidelines [83,84]. It also helps find post-procedural complications. In centers with significant procedural experience, procedures can be performed using ultrasound guidance even in the absence of pleural fluid [85].

In the diagnostic pathway of MPM, ultrasound findings are non-specific but may suggest malignancy when features such as thickening >1 cm and diaphragmatic or chest wall invasion are seen [86]. Nevertheless, definitive diagnosis requires histological confirmation, and TUS should be regarded as a complementary tool rather than a standalone diagnostic modality [29].

Additionally, TUS enables dynamic evaluation of diaphragmatic mobility, with clinical value in both symptom assessment and planning pleural interventions. Reduced diaphragmatic excursion or its inversion could be related to the presence of dyspnea in patients with malignant pleural effusion [87,88], although correlation with symptom relief after thoracentesis may be poor [89,90]. Quantitative assessment of diaphragmatic motion using M-mode ultrasound provides an accessible, bedside measure that complements clinical evaluation and may help in monitoring functional improvement following thoracentesis [91]. In addition, from a procedural standpoint, correct identification of the diaphragmatic position during both inspiration and expiration intuitively reduces the risk of iatrogenic injury during pleural aspiration, drain insertion, or thoracoscopic access. 

An added strength of TUS is its bedside availability, which allows safe evaluation in frail or hospitalized patients for whom transport to the radiology suite may be challenging [92]. Its lack of ionizing radiation further enhances its applicability in repeated and post-procedural assessments [93].

### 5.3. Computed Tomography

Chest CT remains the cornerstone imaging modality in the diagnostic and staging evaluation of MPM. It provides superior anatomical resolution compared with CXR and TUS, enabling detailed assessment of pleural thickening, nodularity, mediastinal involvement, and raises suspicion for chest wall or diaphragmatic invasion [94]. Contrast-enhanced CT evaluation of a venous phase is particularly valuable for differentiating pleural thickening from adjacent structures and for delineating tumor extension along fissures or into extrapleural fat [95].

Characteristic diagnostic CT features of MPM include circumferential or rind-like pleural thickening, contraction of the ipsilateral hemithorax, involvement of the mediastinal pleura, nodularity exceeding 1 cm in size (see Figure 1), and irregular diaphragmatic surface involvement [94,96]. While these findings raise diagnostic suspicion, the CT sensitivity for distinguishing MPM from other pleural malignancies is still limited, underscoring the need for histological confirmation.

Beyond diagnosis, CT plays an essential role in staging according to the most recent 9th TNM classification [97], evaluating disease thickness as sum of maximum pleural thickness (Psum) and maximal fissural thickness (Fmax); local invasion of lung parenchyma, diaphragm, chest wall, pericardium, and mediastinum, as well as detecting nodal and extrathoracic spread [38,98]. It confirms the presence of pleural effusion and characterizes its distribution in the pleural cavity, even if it does not optimally define the degree of septation [99]. The use of CT also helps image-guided pleural biopsies, identifying reachable targets [100].

Nevertheless, CT has limitations: it is relatively insensitive in detecting early disease, small-volume pleural deposits, and peritoneal or pericardial extension [101,102]. Recent developments, such as delayed-phase enhancement [98], dual-energy CT [99], and quantitative techniques, may improve tissue characterization, though their role in routine practice remains under investigation [103,104].

### 5.4. Positron Emission Tomography

18F-fluorodeoxyglucose positron emission tomography/computed tomography (18F-FDG PET/CT) is widely used in the diagnostic pathway of MPM to refine staging and to detect extrathoracic disease that may preclude radical therapy [105]. Compared with anatomical imaging alone, PET/CT provides a whole-body metabolic survey, helping identify unsuspected nodal and distant metastases and assisting TNM staging [104].

Inflammation and prior talc pleurodesis can generate false-positive pleural uptake, whereas low-grade epithelioid lesions or small-volume disease may be falsely negative [104,105]. Recent consensus statements recommend PET/CT primarily for baseline staging and treatment planning, rather than as a stand-alone diagnostic test [104].

Typical PET/CT patterns in MPM include diffuse, rind-like pleural 18F-FDG uptake that tracks along fissures and diaphragmatic surfaces (see Figure 2); nevertheless, pleural metastases from lung or breast adenocarcinoma and certain lymphomas can mimic this pattern [106]. Awareness of pleural PET pitfalls, such as physiological diaphragmatic and myocardic activity, post-procedural inflammation, and pleuritis, is essential for interpretation [106]. Prognostic studies suggest that semi-quantitative parameters (e.g., SUVmax, metabolic tumor volume) may stratify risk and support response assessment, although standardization and prospective validation remain areas of active research [107,108].

PET-CT may also serve as a guide for planning ultrasound-guided biopsy, identifying the most metabolically active and least necrotic sites of disease. However, in cases of prior inconclusive biopsies, its role may be less advantageous, as shown in the TARGET trial, which showed that PET-CT-guided pleural biopsy, compared with CT-guided biopsy, did not reliably increase diagnostic sensitivity [109].

Beyond 18F-FDG, fibroblast activation protein inhibitor (FAPI) tracers (e.g., 68Ga-FAPI) show promising pleural tumor-to-background contrast in MPM, with early prospective and retrospective series suggesting superior lesion conspicuity versus FDG and close association with histologic FAP expression; these agents may enhance detection and staging, but remain investigational pending larger, multi-center trials [110,111,112]. Systematic reviews support a growing role for personalized PET imaging in MPM targeting tumor microenvironment and stromal components, yet most novel tracers are not ready for routine clinical translation [104,113].

### 5.5. Magnetic Resonance Imaging

Magnetic resonance imaging (MRI) is primarily a problem-solving modality in MPM, reserved for situations in which a more exact T-stage definition would alter management [104,114]. Consensus guidance from the International Mesothelioma Interest Group emphasizes MRI’s value as an adjunct to CT for local staging, recommending targeted protocols (T1- and T2-weighted fast spin echo, fat-suppressed sequences, contrast-enhanced T1, and diffusion-weighted imaging) when precise assessment of soft-tissue planes is required [104]. Historical and contemporary data show that MRI can outperform CT for detecting chest wall and diaphragmatic muscle invasion and delineating tumor along the costophrenic recess and fissures, supporting surgical planning and multidisciplinary decision-making [77,114].

Beyond anatomic delineation, diffusion-weighted MRI (DWI) may help differentiate malignant disease from benign pleural plaques and enhance the visual conspicuity of pleural nodularity. This technique increases lesion-to-background contrast by exploiting differences in water diffusivity, such that malignant pleural nodules, characterized by high cellularity and restricted diffusion, appear hyperintense on DWI and hypointense on apparent diffusion coefficient (ADC) maps [115,116]. Consequently, subtle or multifocal pleural nodules, particularly those located along fissures, the diaphragm, or the costophrenic recess, are more readily detected than with conventional MRI or CT alone [116].

MRI-derived tumor volumetry can be larger and more reproducible than CT measures and has been independently associated with survival, suggesting a potential role in response assessment and prognostication; nevertheless, standardization and access remain limiting factors [114]. 

Practical constraints of MRI include availability, longer acquisition times, motion artefacts (respiratory and cardiac), contraindications (implants and claustrophobia), and the need for thoracic ability in acquisition and interpretation. Consistent with guideline recommendations, MRI should be considered in patients in whom differentiating T-stage would change management, rather than for routine imaging [29,104].

## 6. Sampling Techniques

### 6.1. Thoracentesis and Chest Drainage

Thoracentesis is the most common initial invasive procedure in patients with suspected pleural malignancy, providing both symptomatic relief and material for cytological evaluation. Cytological analysis of pleural fluid is readily accessible, but its diagnostic yield for MPM is low [117]. Reported sensitivity ranges from 6% to 45% [117,118,119,120], as distinguishing malignant mesothelial cells from reactive mesothelial hyperplasia often exceeds the capabilities of conventional cytology. In addition, the low level of cellular exfoliation in the pleural space, the subtle cytomorphological features of early disease and the inability to assess stromal invasion make it even harder to reach a final diagnosis without more tests [121]. Cell block preparations may be employed to preserve architecture and facilitate immunohistochemical staining to rule out metastatic carcinomas and reactive mesothelial hyperplasia [122]. Despite the limitations mentioned above, the current literature agrees that a cytological diagnosis of MPM can be established in an appropriate clinical-radiological setting by an experienced pathologist using ancillary tests [122,123].

Cytologic diagnoses reported as “atypical mesothelial proliferation” or “suspicious for mesothelioma” should direct further invasive testing and inform early multidisciplinary discussion, while negative or inconclusive findings on an exudative effusion should prompt histological sampling via image-guided biopsy or medical thoracoscopy [124]. Despite the difficulties, cases diagnosed solely on cytological evaluation achieve earlier therapeutic initiation than those requiring histopathological confirmation [125].

Contemporary guidelines emphasize that thoracentesis should be performed not only for initial fluid analysis but also to assess the symptomatic impact of effusion drainage and to guide subsequent pleural interventions, such as indwelling catheter placement or pleurodesis [124].

Pleural effusion sampling using a chest drain may allow retrieval of larger volumes of fluid. On the other hand, complete lung expansion, when attainable, may induce adhesion formation and impede further diagnostic procedures requiring access to the pleural cavity, such as medical thoracoscopy [126,127]. For this reason, the use of chest drainage as a first diagnostic step should be reserved for patients in whom the intervention is intended to be palliative, with a higher priority on symptom control and pleurodesis than on reaching a conclusive diagnosis.

In summary, thoracentesis and chest drainage remain indispensable first-line procedures in the diagnostic pathway of MPM, but cytological evaluation of pleural fluid must be interpreted with caution and always integrated with clinical and radiological findings.

### 6.2. Percutaneous Pleural Biopsy

Percutaneous pleural biopsy is a central diagnostic tool in MPM, particularly when initial cytological evaluation is inconclusive. The British Thoracic Society (BTS) guidelines recommend that pleural biopsies should always be performed under image guidance (ultrasound or CT), while blind techniques are discouraged due to their low diagnostic performance and higher risk of adverse events [83]. While ultrasound guidance may be readily available to the pulmonologist, CT guidance is often used by radiologists or by pulmonologists with specific training and access to the equipment during limited periods of time. 

The advantage of image-guided pleural biopsy is the ability to obtain targeted samples from focal or nodular pleural thickening, particularly when pleural effusion is minimal or absent. The reported sensitivity is 80–88% for CT-guided and 40–83% for TUS-guided biopsies [100,128,129].

Large (16-18G) needle biopsy tools are used to sample pleural targets, with larger needle sizes correlating with better diagnostic performance [129].

The results may be variable, depending on lesion characteristics and operator expertise. Diagnostic accuracy also depends on pleural thickness, lesion size, and anatomical location. Thin or smooth pleural thickening (<5 mm) significantly reduces the likelihood of obtaining adequate tissue, whereas nodular lesions ≥10 mm are associated with the highest diagnostic yield. Parietal pleural sites are most accessible, while diaphragmatic and mediastinal locations carry greater procedural risk and lower yield due to limited needle access [128]. 

Image-guided techniques are minimally invasive and have low complication rates (3–7%). However, their small tissue samples may be insufficient for full histologic characterization or confirmation of stromal invasion, which remains essential for definitive diagnosis [71,130,131].

### 6.3. Medical Thoracoscopy

Medical thoracoscopy (MT), also called local anesthetic thoracoscopy or pleuroscopy, combines direct visualization of the pleural space with the ability to obtain large, targeted biopsies under direct vision and to prevent malignant fluid re-accumulation. The BTS guideline recommends MT as a first-line diagnostic approach in patients with an undiagnosed unilateral pleural effusion when malignancy is suspected, provided adequate services are available [124].

MT provides the highest diagnostic performance among medical-performed sampling techniques for pleural malignancy [132]. Reported sensitivity for detecting pleural malignancy, including MPM, is 92–95%, with specificity approaching 100%, and an overall diagnostic yield consistently above 90–95% [133,134,135]. 

Semirigid thoracoscopy may be an alternative to rigid MT. Despite randomized pilot studies showing equal diagnostic performance for exudative pleural effusions [135,136], it is reasonable to expect that rigid MT may perform better on large, hard lesions, such as the pleural thickening that is frequently found in MPM [136,137,138]. To overcome the limitation of the sample size obtained with forceps, pleural cryobiopsy was introduced, but with less promising results [139]. A practical alternative for centers working with semirigid instrumentation would be to perform a second parietal thoracoscopic access to introduce a pair of rigid forceps, taking samples under the endoscopic view of the semirigid thoracoscope [140].

Narrow band imaging (NBI), available for semirigid models, enhances pleural visualization during medical thoracoscopy by using specific light wavelengths (415 and 540 nm) to highlight vascular patterns and microstructural changes [141,142]. In MPM, NBI may reveal irregular, dilated, and tortuous vessels, improving the distinction between malignant and benign pleural areas and enabling more precise biopsy targeting (see Figure 3).

An added strength of MT lies in its ability to directly recognize features of MPM staging, including fissural, diaphragmatic, and pericardial involvement. However, these intra-procedural findings are not a substitute for formal radiologic TNM staging, but they provide immediate and valuable insights that can guide multidisciplinary team discussions and prompt the need for complementary advanced imaging (MRI, PET-CT) [143]. 

MT is a safe procedure with a low complication rate (2–7%) and mortality below 0.1% when performed by experienced operators [124,144]. The most frequent adverse events are prolonged air leak (1–2%), minor bleeding (<1%), and pleural infection (1%), while transient pain, post-procedural fever, or subcutaneous emphysema are common but self-limiting [144]. Severe hemorrhage, cardiopulmonary events and lung lacerations are rare [145]. Tumor tract seeding occurs more often in MPM than in other histotypes, reaching up to 26% [146,147]. Overall, MT has an excellent safety profile, particularly when standard procedural protocols and infection-control measures are observed.

Rapid on-site evaluation (ROSE) has been proposed during MT sampling to accelerate the diagnosis of malignant pleural effusion, with significant positive results that, unfortunately, have not yet been integrated into global clinical practice [148,149,150]. However, MPM may be a setting in which such an application may not add significant value, given the difficulty of diagnosing cytologically alone without additional tests such as IHC.

MT also offers the advantage of therapeutic talc poudrage pleurodesis in the same sitting, which is recommended when recurrent malignant pleural effusion is present and the lung is deemed sufficiently expandable [151,152].

Although MT may be considered the gold standard for medical diagnosis, where expertise and facilities are available, image-guided percutaneous biopsy also has high sensitivity [153,154]. The latter is a safer, less invasive alternative for patients who are unsuitable for MT [154] and who exhibit large pleural nodules or thickening accessible via the percutaneous route [155]. The two sampling methods probably have different ideal application cohorts. For example, MT cannot be performed in the absence of an accessible pleural cavity, which is a condition that can occur in cases of MPM that manifests only with pleural thickening, without pleural effusion and with fixed adhesion between the visceral and parietal pleura (see Figure 4). These techniques should not be compared but rather considered two useful, interchangeable tools in the pulmonologist’s hands.

### 6.4. EBUS-TBNA and EUS-B

A few isolated reports have described the use of endobronchial ultrasound-guided transbronchial needle aspiration (EBUS-TBNA) and endoscopic ultrasound with a bronchoscope-guided fine-needle aspiration (EUS-B-FNA) in the diagnostic and staging work-up of MPM [156]. Although the amount of tissue obtained from pleural lesions remains limited with these approaches, both techniques enable minimally invasive sampling of the mediastinal pleura and mediastinal and hilar lymph nodes located near the central airways or esophagus [157,158,159]. Their primary utility lies in nodal staging, particularly in distinguishing N1 from N2 disease, with significant therapeutic implications.

In a published retrospective evaluation, EBUS-TBNA achieved a diagnostic sensitivity of 59% and a negative predictive value of 57% in MPM [157]. EUS-B-FNA extends the sampling range to paraoesophageal and subdiaphragmatic nodes, improving access to nodal stations not reachable by EBUS alone [159]. Thus, despite their diagnostic performance not being ideal, these endosonographic modalities may be considered minimally invasive first-line procedures for pleural or mediastinal evaluation in patients with suspected or confirmed MPM, before considering surgical alternatives or when MT or percutaneous sampling may not be possible.

There are no current reports on the application of cryobiopsy or intranodal or intralesional forceps biopsy in this setting.

## 7. Histological Classification

### 7.1. Different Histotypes and Their Meaning

The 2021 WHO Classification of Tumors of the Pleura described the three principal histological subtypes of diffuse pleural mesothelioma—epithelioid, sarcomatoid, and biphasic—while integrating new morphologic refinements [53] (see Figure 5). Within epithelioid mesothelioma, the classification emphasizes architectural, cytologic, and stromal patterns that carry prognostic significance. Favorable histologic features include tubulopapillary, trabecular, or adenomatoid architecture, lymphohistiocytoid cytology, and predominant myxoid stroma [160,161]. Conversely, a solid pattern involving ≥50% of the tumor, micropapillary growth, rhabdoid morphology, pleomorphism, and necrosis are associated with poor prognosis [162,163,164].

The WHO 2021 classification adopts a two-tier grading system for epithelioid mesothelioma, derived from the nuclear grade proposed by Kadota et al. and the multi-institutional validation by Rosen et al., which combines nuclear atypia, mitotic count, and the presence of necrosis to stratify tumors into low- and high-grade categories [164,165]. Tumors exhibiting transitional morphology, intermediate between epithelioid and sarcomatoid forms, are now categorized as sarcomatoid due to their molecular and clinical aggressiveness [166,167,168,169]. Likewise, lymphohistiocytoid features, previously confined to the epithelioid category, can now occur across all histotypes but retain prognostic relevance in sarcomatoid tumors, in which they are associated with improved outcomes [170,171,172].

The new classification formally introduced the definition of mesothelioma in situ (MIS) as a distinct preinvasive entity, reflecting advances in the understanding of MPM progression [53]. MIS is defined by the presence of a single layer of cytologically bland mesothelial cells lining the pleural surface without evidence of stromal invasion, accompanied by loss of BAP1 expression and/or homozygous deletion of CDKN2A detected by fluorescence in situ hybridization (FISH), or methylthioadenosine phosphorylase (MTAP) loss by immunohistochemistry [173,174,175]. Additional clinical criteria for diagnosis include: a non-resolving pleural effusion, no thoracoscopic or imaging evidence of frank tumor appearance, and a multidisciplinary diagnosis reached after proper discussion of each case. Clinically, MIS may precede the development of invasive diffuse pleural mesothelioma by several months or years. Longitudinal studies have shown progression to invasive disease in a subset of cases. Progression is more likely in lesions showing BAP1 loss and/or CDKN2A deletion [53]. The identification of MIS therefore provides an opportunity for earlier diagnosis and subsequent active surveillance in high-risk individuals, especially those with a history of asbestos exposure or germline BAP1 mutation. 

### 7.2. Immunohistochemistry and Molecular Testing

Immunohistochemistry (IHC) and molecular testing have become integral to the differential diagnosis and classification of mesothelioma (Table 1). Among carcinoma markers, Claudin-4 has emerged as a highly reliable discriminator between adenocarcinoma and mesothelioma, with 77–100% sensitivity and 99–100% specificity [176,177,178]. The combination of BAP1 loss and homozygous p16 (CDKN2A) deletion by FISH and/or MTAP loss has become a cornerstone for distinguishing malignant mesothelioma from reactive mesothelial proliferations, while EZH2 overexpression represents an additional promising marker [166,179,180,181,182,183,184,185].

The WHO classification underscores the need for rigorous assay validation and internal controls for these markers, particularly in cytology specimens, where fixation methods can affect antibody performance [186,187,188,189,190]. While retained BAP1 or MTAP expression does not exclude mesothelioma, the combined loss of both strongly supports malignancy. From a genomic perspective, diffuse pleural mesothelioma exhibits recurrent mutations in BAP1, NF2, TP53, SETD2, SETDB1, and CDKN2A [191,192]. These alterations not only aid diagnosis but also define emerging biologic subgroups with prognostic relevance.

## 8. Staging and Multidisciplinary Evaluation

### 8.1. Current Staging System and Its Clinical Implications

The ninth edition of the International Association for the Study of Lung Cancer (IASLC) TNM classification for MPM introduces significant refinements designed to improve prognostic stratification and clinical applicability compared with the eighth edition. The most substantial update concerns the T descriptors, which now integrate quantitative measurements of pleural thickness (Psum) across three axial levels and along fissures (Fmax), replacing the previous qualitative definitions of pleural involvement. The new T categories distinguish tumors limited to the ipsilateral pleura (T1) from those showing increasing pleural thickness, mediastinal fat invasion, or focal soft-tissue extension (T2–T3), while T4 denotes bony, mediastinal organ, or contralateral pleural invasion [193]. These objective criteria address one of the main limitations of the eighth edition: its lack of reproducibility between clinical and pathological staging.

The N classification retains the structure introduced in the eighth edition, differentiating N1 (ipsilateral intrathoracic) from N2 (contralateral or supraclavicular) disease, as these categories have been shown to correlate distinctly with survival without requiring further subdivision [97]. The M descriptor remains unchanged, with M1 denoting any distant metastasis. Together, these descriptors yield simplified stage groupings: stages I–IIIA correspond to T1–T3 with N0–2 disease, stage IIIB encompasses any T4 without distant metastasis, and stage IV includes M1 involvement [194].

Clinically, these refinements facilitate a more standardized and measurable approach to staging through imaging and biopsy, enhance reproducibility across institutions, and allow closer alignment between clinical and pathological staging. Significantly, they improve patient selection for multimodal therapy, while providing a platform for integrating molecular and radiomic markers in forthcoming iterations of the staging system [97,193,194].

### 8.2. The Role of the Multidisciplinary Team

The IASLC ninth edition TNM classification emphasizes the pivotal role of the multidisciplinary team (MDT) in achieving accurate staging and optimized treatment planning. Given the increased complexity of the revised descriptors, particularly the quantitative T definitions and the integration of radiological and pathological parameters, collaboration among radiologists, pulmonologists, thoracic surgeons, oncologists, and pathologists is essential for harmonizing clinical and pathological staging [74,195]. The MDT ensures consistent interpretation of imaging-based measurements, appropriate tissue sampling planning, and evidence-based selection for multimodality therapy or clinical trial enrollment, thereby enhancing prognostic accuracy and improving outcomes across institutions [97,193,194].

## 9. Emerging Diagnostic Approaches and Biomarkers

### 9.1. Liquid Biopsies

Liquid biopsy is a promising non-invasive approach for the detection and monitoring of MPM. Circulating tumor DNA (ctDNA), microRNAs, and methylation signatures have shown potential for early diagnosis, minimal residual disease assessment, and monitoring of treatment response [29,196]. Despite encouraging preliminary data, sensitivity is still variable due to the typically low tumor burden and low circulating DNA levels in MPM. Current BTS guidelines highlight liquid biopsy as an adjunctive rather than standalone diagnostic tool pending further standardization and validation in large prospective cohorts [29]

### 9.2. Novel Biomarkers

Beyond conventional immunohistochemical markers, several soluble biomarkers, including fibulin-3, soluble mesothelin-related peptides (SMRP), and osteopontin, have been proposed as diagnostic or prognostic tools [29,197]. Although these markers may aid risk stratification and disease monitoring, variability in assay performance limits their clinical value in isolation. The BTS guideline recommends that no circulating biomarker should replace tissue diagnosis due to low sensitivity and specificity, but supports their use in testing patients with suspicious cytology who are not fit enough for more invasive diagnostic tests [29]. Integration of multi-omic panels combining genomic, proteomic, and epigenetic features may, in the future, enhance diagnostic accuracy [197].

### 9.3. Advances in Artificial Intelligence and Radiomics

Artificial intelligence (AI) and radiomics are increasingly applied to thoracic imaging to extract quantitative features beyond human perception. Machine-learning algorithms trained on CT and PET-CT datasets have demonstrated the ability to differentiate MPM from metastatic pleural disease, predict histological subtype, and perform advanced automated volumetric assessment [198,199,200,201,202]. Li et al. reported diagnostic accuracies above 84% using CT-based machine-learning models to distinguish MPM from metastatic pleural tumors [203]. Similarly, convolutional neural networks (CNNs) have shown potential to standardize radiological interpretation and reduce inter-observer variability, paving the way for AI-assisted integration into future diagnostic and staging systems [204,205,206]. However, these models rely on limited datasets and lack standardized validation frameworks. Thus, AI-based tools should still be regarded as investigational and, if used, applied only as adjuncts to expert multidisciplinary assessment until robust, prospective, multicenter validation and clear regulatory standards are established.

### 9.4. Other Technologies

Non-invasive diagnostic tools such as volatile organic compound (VOC) breath analysis are emerging as promising adjuncts for MPM detection. VOCs, measurable by gas chromatography–mass spectrometry (GC–MS) [207] or electronic nose (eNose) [208] technologies, reflect tumor-related metabolic and inflammatory changes. Several studies have shown that VOC profiling can differentiate MPM from asbestos-exposed and healthy individuals with reported accuracies ranging from 80–90% or higher [207,209,210]. Recent advances integrating sensor arrays and machine-learning algorithms have improved reproducibility and reduced analytical variability [211]. While breathomics may ultimately enable early detection and risk monitoring, further validation and methodological standardization are needed before clinical implementation.

## 10. Communicating the Diagnosis

Delivering a diagnosis of MPM is a necessary step at the end of the diagnostic process, and the physician responsible for this stage is often the pulmonologist. It requires a sensitive, structured, and patient-centered approach. The disease’s aggressive nature, poor prognosis, and frequent occupational etiology make disclosure uniquely complex, often evoking feelings of shock, guilt, anxiety, and anger in both patients and families [212]. Effective communication must therefore extend beyond conveying clinical facts to encompass emotional, social, and ethical dimensions of care.

A multidisciplinary consultation, ideally involving the pulmonologist, oncologist, and palliative-care professional, allows information to be presented gradually and consistently, with sufficient time for questions and shared decision-making [213]. Patients benefit when discussions focus on clear explanation of the disease stage, available treatments, prognosis, and supportive-care pathways, while acknowledging uncertainty and validating emotional reactions [213].

The impact on the patient’s caregivers is equally profound: recent qualitative research highlights significant psychological distress, anticipatory grief, and caregiver burden, often intensified by witnessing rapid clinical decline and navigating compensation or legal processes related to MPM’s diagnosis [211,213,214]. Early psychosocial support, continuity of care, and access to dedicated counselling have been shown to mitigate distress for both patients and families [213,215]. Ultimately, effective communication improves satisfaction with care and supports adaptation to this life-limiting diagnosis [213].

## 11. Conclusions and Future Directions

Despite notable progress in imaging, pathology, and molecular profiling, the diagnostic pathway for MPM continues to face challenges, particularly in achieving earlier and more accurate detection. Early detection of MPM remains both a conceptual and ethical dilemma. Because the disease progresses insidiously and becomes symptomatic only at an advanced stage, earlier diagnosis—whether through incidental imaging findings or surveillance of asbestos-exposed individuals—often merely extends the interval between diagnosis and death, not continuously improving overall survival. This phenomenon illustrates lead-time bias. Likewise, length-time bias favors the identification of indolent tumors, while aggressive ones tend to become symptomatic and are diagnosed at a later stage. This raises an ethical question: Is it justifiable to diagnose a patient early when no curative treatment exists? Early diagnosis can induce anxiety and social stigma, and expose patients to invasive diagnostic procedures without consistent therapeutic benefit, despite legal recognition of the disease. However, despite all these considerations, a subset of patients with epithelioid histology and early-stage MPM may benefit from symptom control and multimodal therapy, and may also be eligible for clinical trials [216,217,218].

Ongoing trials reflect a growing trend toward less invasive, precision-based approaches, including NCT06790082, which evaluates FAPI PET-CT for improved imaging specificity, and NCT01950572, which focuses on tissue biobanking and biomarker validation. These studies highlight the movement toward integrated diagnostic frameworks that combine imaging, molecular, and computational tools. Future progress will depend on standardizing diagnostic algorithms, expanding multicenter collaborations, and embedding AI-driven image and data analysis into clinical workflows. Ultimately, innovation in MPM diagnosis must aim to reduce invasiveness and improve patient stratification, laying the groundwork for a precision medicine era in mesothelioma care.

## Figures and Tables

**Figure 1 cancers-17-03866-f001:**
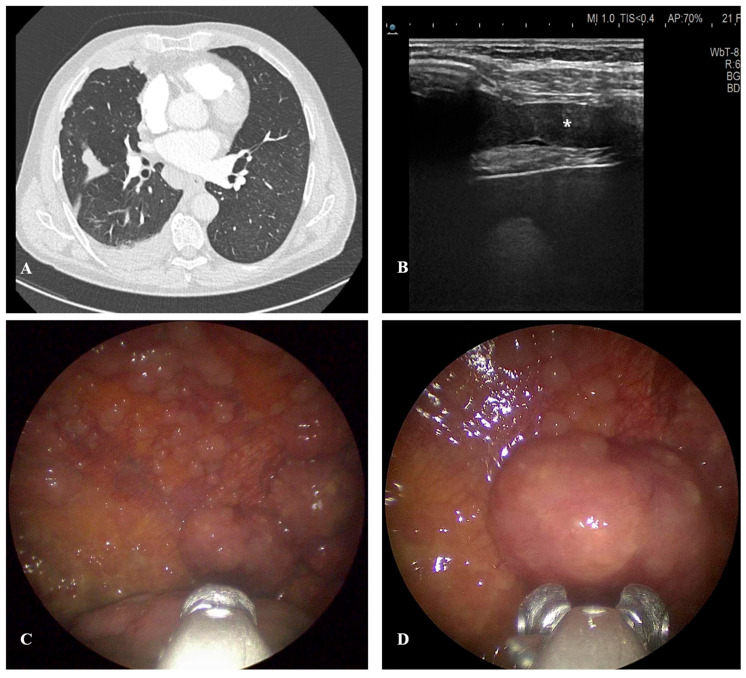
Case of malignant pleural mesothelioma. (**A**) CT scan (slice thickness 1.25 mm) showing diffuse pleural nodules affecting the parietal, visceral and fissural pleura. (**B**) Ultrasound scan (linear probe, maximum depth of 60mm) of the costophrenic sinus showing evidence of pleural thickening (identified by the symbol ⁎). (**C**) inspection of the pleural cavity by medical thoracoscopy showing diffuse nodules on the parietal pleura. (**D**) sampling of a lesion by rigid medical thoracoscopy. CT: computed tomography.

**Figure 2 cancers-17-03866-f002:**
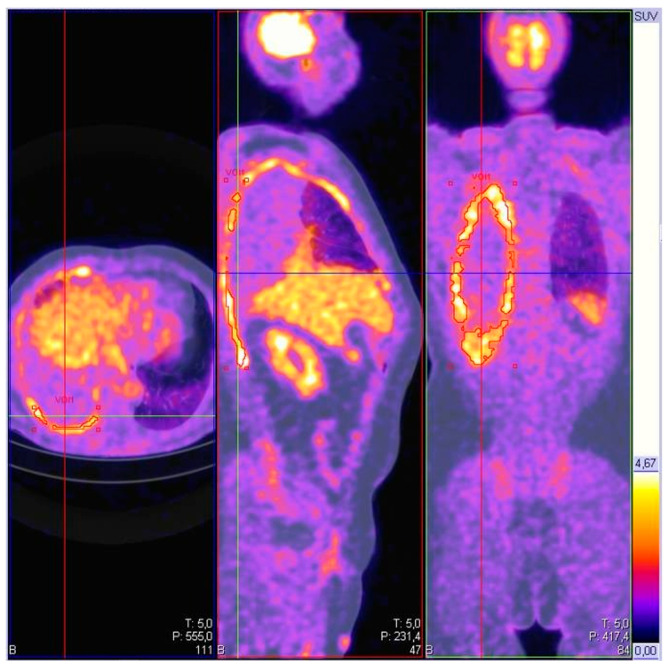
PET-CT evaluation with 18F-FDG of a case of malignant pleural mesothelioma. Typical rind-like appearance of the mesothelioma extension on the pleural surface—axial, sagittal and coronal evaluation. SUV max: 12.88.

**Figure 3 cancers-17-03866-f003:**
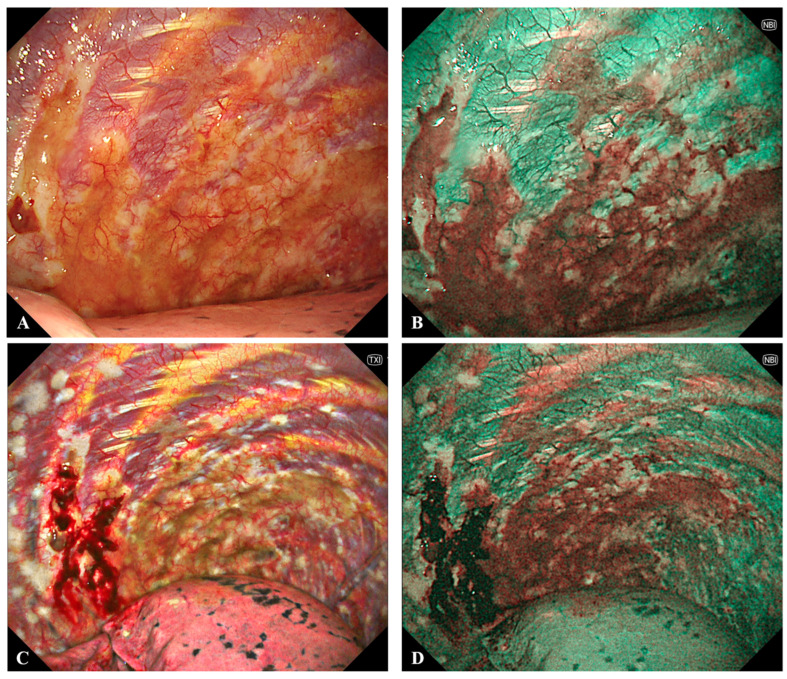
Case of malignant pleural mesothelioma, evaluation using semi-rigid medical thoracoscopy and narrow band imaging (NBI). (**A**) Area of asbestos-related pleuritis with fibrohyaline plaques under white light, where no additional pathological changes can be identified with certainty. (**B**) NBI appearance of the pathological area (in red) clearly demarcates the region with mesothelioma infiltration, characterized by increased microvascularization and alterations in the parietal pleural surface. (**C**) Area of asbestos-related pleuritis with fibrohyaline plaques (white) under Texture and Color Enhancement Imaging (TXI) after biopsy. (**D**) Using the NBI technique clearly shows the area of mesothelioma infiltration (in red).

**Figure 4 cancers-17-03866-f004:**
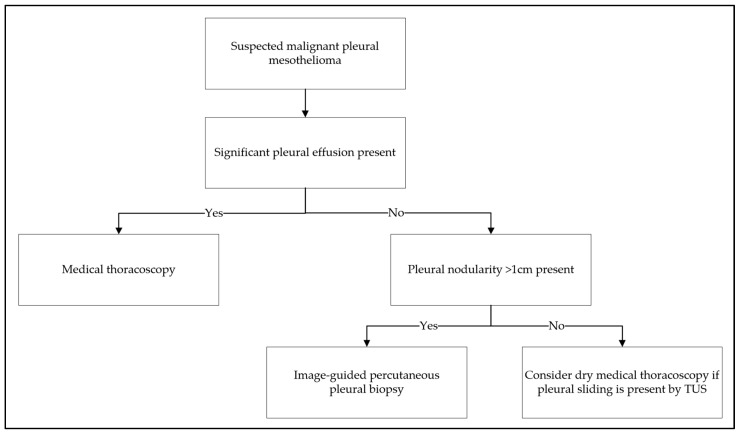
Proposed diagnostic algorithm for malignant pleural mesothelioma guiding the choice between medical thoracoscopy and imaging-guided percutaneous biopsy in the diagnostic pathway. TUS: thoracic ultrasound.

**Figure 5 cancers-17-03866-f005:**
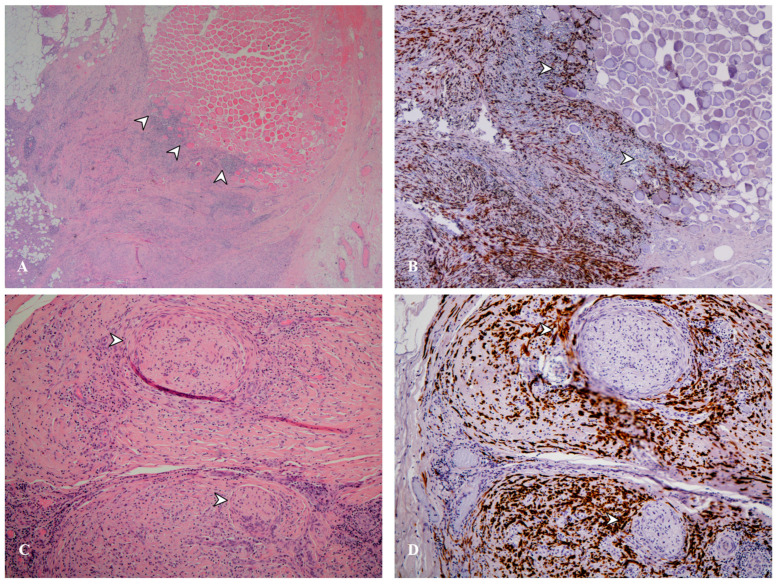
Case of malignant pleural mesothelioma with sarcomatoid histotype. (**A**) histological section at 20× magnification, stained with hematoxylin and eosin showing invasion (arrowhead) of the skeletal muscle of the chest wall. (**B**) histological section at 40× magnification, stained with cytokeratin CAM 5.2 showing invasion (arrowhead) of the skeletal muscle of the chest wall. (**C**) histological section at 100× magnification, with hematoxylin and eosin staining showing perineural invasion (arrowhead). (**D**) histological section at 100× magnification, with CAM 5.2 cytokeratin staining showing perineural invasion (arrowhead).

**Table 1 cancers-17-03866-t001:** Immunohistochemical panels associated with malignant pleural mesothelioma and some differential diagnoses.

Histotype	Markers
Epithelioid malignant pleural mesothelioma	calretinin/WT1/CK5/CK6/D2-40 (≥2 strongly positive); BAP1 loss, MTAP loss, p16/CDKN2A homozygous deletion, GLUT-1, membranous EMA
Lung adenocarcinoma	TTF-1/Napsin A/Claudin-4/MOC-31/Ber-EP4 and CEA positive; mesothelial markers negative or focal; BAP1/MTAP retained, no p16 homozygous deletion, GLUT-1 variable
Metastatic breast carcinoma	ER/PR/GATA3/mammaglobin/GCDFP-15 positive; mesothelial markers negative; no p16 homozygous deletion; BAP1/MTAP retained
Reactive mesothelial proliferation	mesothelial markers are positive; BAP1 and MTAP retained, no p16 homozygous deletion, GLUT-1 negative/weak, low Ki-67
Sarcomatoid malignant pleural mesothelioma	keratin-positive, usually mesothelial markers negative, lineage markers (p40/p63, TTF-1, etc.) may help; BAP1 often retained

## Data Availability

Not applicable.

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
