# Peer review of "Pleural Mesothelioma Diagnosis for the Pulmonologist: Steps Along the Way"

_cancers, 2025, doi:10.3390/cancers17233866_

Round 1
Reviewer 1 Report
Comments and Suggestions for Authors
Dear Authors,
Thank you for submitting your manuscript "Pleural Mesothelioma Diagnosis for the Pulmonologist: Steps Along the Way" to Cancers. As the reviewer, I find it a useful overview but there are some comments on your manuscript:
- Expand methodology details on deduplication, screening, and bias mitigation in Evidence Review Accelerator.
- Revise the epidemiology section's male-female ratio with regional variability.
- Enhance cytology discussion with liquid biopsy technique.
- Critically evaluate AI limitations and cite more prospective studies.
- Balance pessimism on early detection biases by noting benefits like trial eligibility and symptom management.
- Add tables: imaging modalities (sensitivity/specificity); IHC panels for differential diagnosis. Include a summary table comparing modalities.
- Provide a clear diagnostic algorithm in Section 6, clarify when to choose medical thoracoscopy or image-guided biopsy based on patient factors.
- Expand the ethical dilemma on early diagnosis into a dedicated subsection earlier in the paper.
- Conduct a thorough English edit for grammar and clarity.
Best regards
Author Response
Please also see the cover letter.
Reply to reviewer 1:
1. Expand methodology details on deduplication, screening, and bias mitigation in Evidence Review Accelerator.
The following paragraph was added to the Methodology section: “Following completion of the literature searches, all retrieved records were im-ported into the Evidence Review Accelerator (https://tera-tools.com/), which facilitated identification and removal of both exact and near-duplicate entries. Subsequently, two reviewers independently screened titles and abstracts to identify eligible documents. Full-text articles deemed potentially pertinent were evaluated in the same independent fashion. Discrepancies between reviewers were addressed through structured discus-sion, with recourse to a third reviewer when consensus could not be reached.”
Bias was mitigated during the references initial review phase, but a degree of bias induced by individual references proposed by individual authors in various sections of the manuscript text is inevitable.
2. Revise the epidemiology section's male-female ratio with regional variability.
The paragraph was modified: “The male-to-female ratio for MPM prevalence is usually 4:1, with possible significant interregional variations worldwide”
3. Enhance cytology discussion with liquid biopsy technique.
The discussion on liquid biopsy has been moved to section 9, “Emerging diagnostic approaches and biomarkers,” after initially being considered for inclusion in the cytology section, as it is still significantly underutilized and has only recently been introduced, even when compared to the routine of lung cancer, for example.
4. Critically evaluate AI limitations and cite more prospective studies.
The following addition has been made to section 9.3: “Accordingly, AI-based tools should still be regarded as investigational and, if used, applied only as adjuncts to expert multidisciplinary assessment until robust, prospective, multicenter validation and clear regulatory standards are established.”
5. Balance pessimism on early detection biases by noting benefits like trial eligibility and symptom management.
The paragraph was modified: “Despite notable progress in imaging, pathology, and molecular profiling, the di-agnostic pathway for MPM continues to face challenges, particularly in achieving earlier and more accurate detection. Early detection of MPM remains both a conceptual and ethical dilemma. Because the disease progresses insidiously and becomes symptomatic only at an advanced stage, earlier diagnosis - whether through incidental imaging findings or surveillance of asbestos-exposed individuals - often merely extends the in-terval between diagnosis and death, not continuously improving overall survival. This phenomenon illustrates lead-time bias. Likewise, length-time bias favors the identifi-cation of indolent tumors, while aggressive ones tend to become symptomatic and are diagnosed at a later stage. This raises an ethical question: Is it justifiable to diagnose a patient early when no curative treatment exists? Early diagnosis can induce anxiety and social stigma, and expose patients to invasive diagnostic procedures without consistent therapeutic benefit, despite legal recognition of the disease. However, despite all these considerations, a subset of patients with epithelioid histology and early-stage MPM may benefit from symptom control and multimodal therapy, and may also be eligible for clinical trials”
6. Add tables: imaging modalities (sensitivity/specificity); IHC panels for differential diagnosis. Include a summary table comparing modalities.
A table (Table 1) concerning differential diagnosis by immunohistochemistry has been added. The authors believe that additional tables could overload the text.
7. Provide a clear diagnostic algorithm in Section 6, clarify when to choose medical thoracoscopy or image-guided biopsy based on patient factors.
Figure 4 has been added with a related diagnostic algorithm.
8. Expand the ethical dilemma on early diagnosis into a dedicated subsection earlier in the paper.
The authors chose to summarize the concluding paragraph and the subsection dedicated to the two biases rather than adding another section to the text, so as not to overload the reader.
9. Conduct a thorough English edit for grammar and clarity.
The text has been reviewed by several native English speakers.

Reviewer 2 Report
Comments and Suggestions for Authors
This article is a narrative review devoted to the epidemiology, risk factors, pathogenesis, and diagnostic approaches to pleural mesothelioma. Particular attention is paid to the role of the pulmonologist in the diagnostic process and management of patients with mesothelioma. The paper is written very clearly and concisely, and well illustrated.
I have no comments or concerns.
My only suggestion is to add an MRI image to the article.
Author Response
Please also see the cover letter.
We thank the reviewer. We believe that the paper already contains sufficient illustrations and that any further additions would overwhelm the reader.

Reviewer 3 Report
Comments and Suggestions for Authors
This review accurately describes the diagnostic procedures of mesothelioma, a complex disease with poor prognosis and considerable social and legal implications. Just a few concerns to better accompany the potential reader in this very precise and comprehensive review.
- Line 36: “narrative review”. Line 40: studies “were systematically identified.”
Is this a narrative or systematic review?
- In Figure 4, the perineural invasion has been illustrated and described (not mentioned in the text). Do the authors think that perineural invasion plays a particular importance in the natural history or prognosis of mesothelioma?
Author Response
Please also see the cover letter
Reply to reviewer 3:
We thank the reviewer. The review is narrative in nature, with more in-depth bibliographic research than a traditional narrative one, but not with such precision, nor with the assistance of a methodologist, as to make it a systematic review.
Perineural and vascular invasion are histological features that indicate a more aggressive neoplasm. This paper does not aim to focus on the parameters related to the prognosis for the disease, but only on its diagnostic pathway.

Reviewer 4 Report
Comments and Suggestions for Authors
This is a timely, clinically relevant narrative review that aims to summarize the diagnostic pathway for malignant pleural mesothelioma (MPM) with a practical emphasis for pulmonologists. The manuscript covers epidemiology, imaging, sampling techniques, pathology, staging, emerging biomarkers and communication. The scope is appropriate for a clinical review and the literature cited is extensive and recent.
I only have some minor suggestions to improve the manuscript:
- The content is overly extensive, and my only recommendation is to summarize the manuscript to enhance readability.
- The paper presents as a narrative review that claims a systematic search was performed (“English-language studies published between January 2005 and June 2025 were systematically identified from PubMed and Scopus”) but lacks essential methodological transparency and evidence appraisal. The manuscript alternates language (“narrative review, systematic in its literature identification”) that creates confusion. If the authors intended a systematic review, they must follow reporting standards (PRISMA). If it is a narrative review, remove “systematic” claims and be transparent about limitations. Please chose between narrative or systematic review and comprise with the guidelines for the on chosen.
- Figures lack essential information: image acquisition parameters (CT slice thickness, modality and sequences for MRI), scale bars, patient consent/ethics statements for identifiable images, and clear arrows/labels. For histology images, include magnification, and scale. If images are from patients, confirm documented patient consent or institutional approval.
- Include a limitation paragraph related to the results presented.
Author Response
Please also see the cover letter.
Reply to reviewer 4:
1. The content is overly extensive, and my only recommendation is to summarize the manuscript to enhance readability.
Prior to submission, the authors have already summarized the paper extensively and optimized the order in which the concepts are presented. Unfortunately, the literature to be taken into consideration was significantly extensive and cannot be summarized briefly without omitting some topics (another reviewer even pointed out that some sections could be added). Furthermore, better defining the aspects of a specialization (pulmonology) and giving the paper a practical cut made it necessary to extend some sections.
2. The paper presents as a narrative review that claims a systematic search was performed (“English-language studies published between January 2005 and June 2025 were systematically identified from PubMed and Scopus”) but lacks essential methodological transparency and evidence appraisal. The manuscript alternates language (“narrative review, systematic in its literature identification”) that creates confusion. If the authors intended a systematic review, they must follow reporting standards (PRISMA). If it is a narrative review, remove “systematic” claims and be transparent about limitations. Please chose between narrative or systematic review and comprise with the guidelines for the on chosen.
The term “systematic” has been removed in reference to this review when mentioned in the text.
3. Figures lack essential information: image acquisition parameters (CT slice thickness, modality and sequences for MRI), scale bars, patient consent/ethics statements for identifiable images, and clear arrows/labels. For histology images, include magnification, and scale. If images are from patients, confirm documented patient consent or institutional approval.
We thank the reviewer for their careful comments. The definition of slice thickness has been added to the chest CT images. For the histological images, magnification details have been added for each photo. The images have been cropped both to maximize formatting and to ensure patient anonymity. Where indicated, arrowheads have been added to indicate the elements to which the notes refer. Informed consent for the use of images was obtained from all patients for all radiological and pathological images (see section “Informed Consent Statement: Informed consent was obtained from all patients included in the clinical cases described.”).
We would like to thank all the reviewers for taking the time to read and comment on our paper.
